# Association between thrombocytopenia and 180-day prognosis of COVID-19 patients in intensive care units: A two-center observational study

Yuan Zhu[1,2], Jing Zhang[1,2], Yiming Li[1,2], Fang Liu[1,2], Qing Zhou[1]*, Zhiyong Peng[1,2]*

1 Department of Critical Care Medicine, Zhongnan Hospital of Wuhan University, Wuhan, Hubei, China,
2 Clinical Research Center of Hubei Critical Care Medicine, Wuhan, Hubei, China

* Pengzy5@hotmail.com (ZYP); zhouqing680211@126.com (QZ)

## Abstract

### Background

Thrombocytopenia has been proved to be associated with hospital mortality in patients with severe acute respiratory syndrome coronavirus 2 (SARS-CoV-2) infections. However, the detailed association of thrombocytopenia with subsequent progression of organ functions and long-term prognosis in critically ill COVID-19 patients remains to be explored.

### Methods

Medical records of 167 confirmed cases of critically ill COVID-19 from February 16 to March 21, 2020 were collected in this two-center retrospective study. 180-day's outcome and clinical organ development in patients with thrombocytopenia and non-thrombocytopenia were analyzed.

### Findings

Among all 167 patients, the median age was 66 years and 67.07% were male. Significant differences were noticed in laboratory findings including white blood cells, blood urea, total bilirubin, lactate dehydrogenase and SOFA score between groups of thrombocytopenia and non-thrombocytopenia. Older age, lower platelet count and longer activated partial thromboplastin time at admission were determined to be risk factors of 28-day mortality, and all three, together with higher white blood cells were risk factors of 180-day mortality. Subsequent changes of six-point ordinal scale score, oxygenation index, and SOFA score in patients with thrombocytopenia showed marked worsening trends compared with patients without thrombocytopenia. Patients with thrombocytopenia had significantly higher mortality not only in 28 days, but also in 90 days and 180 days. The time-course curves in non-survival group showed a downtrend of platelet count and oxygenation index, while the curve of six-point ordinal scale kept an uptrend. Kaplan-Meier analysis indicated that patients with thrombocytopenia had much lower probability of survival (p<0.01).

**Data Availability Statement:** All relevant data are within the manuscript and its Supporting Information files.

**Funding:** This work was supported by the Special Project for Significant New Drug Research and Development in the Major National Science and Technology Projects of China (2020ZX09201007).

**Competing interests:** The authors have declared that no competing interests exist.

## Interpretation

The thrombocytopenia was associated with the deterioration of respiratory function. Baseline platelet count was associated with subsequent and long-term mortality in critically ill COVID-19 patients.

## Introduction

The ongoing coronavirus disease-19 (COVID-19), which is caused by the infections of severe acute respiratory syndrome coronavirus 2 (SARS-CoV-2), has led to more than 91 million confirmed cases and nearly 2 million deaths globally as of January 11, 2020 [1]. The clinical manifestations in COIVD-19 patients have been extensively reported since the outbreak. Patients with COVID-19 can represent with abdominal symptoms, acute heart injury, secondary infections, acute liver injury, acute kidney injury and coagulation abnormalities in addition to pulmonary symptoms [2–4]. And the main changes in complete blood cells are characterized with lymphopenia and thrombocytopenia [5].

Thrombocytopenia is a common manifestation and also an indicator of poor prognosis of SARS, MERS and COVID-19 according to previous researches [6–9]. Low platelet count has been proven to be related to disease severity and hospital mortality in COVID-19 patients [10–12]. What was noteworthy was that severe COVID-19 patients had significantly lower platelet count than non-severe patients [11, 13]. However, the detailed association between platelets and subsequent organ functions in severe COVID-19 patients in intensive care units (ICUs) remain to be explored. The long-term prognosis of critically ill COVID-19 patients also remain to be uncertain. Therefore, we investigated the subsequent organ dysfunction and 180-day outcome in critically ill COVID-19 with thrombocytopenia.

## Methods

### Study design and participants

The study enrolled ICU patients from 2 centers in Wuhan, namely Zhongnan Hospital of Wuhan University and Leishenshan Hospital, which were managed by the same medical team from Zhongnan hospital since early February of 2020. Adult patients diagnosed with COVID-19 based on the guidance [14] issued by National Health Commission of the People's Republic of China (7th edition), and admitted to the two ICUs from January 5 to March 21 were enrolled in this study. And those with pregnancy or lactation, malignant tumor with a life expectancy of less than 3 months, immunodeficiencies (i.e., acquired immune deficiency syndrome and leukemia), and diseases that cause excessive consumption of platelets (i.e., primary immune thrombocytopenia and heparin induced thrombocytopenia) were all excluded. Clinical outcomes (mortality) were monitored up to September 17, 2020. Ethical approval was obtained from the Medical Ethics Committee of Zhongnan Hospital of Wuhan University (2020057K). Considering the epidemic situation, oral informed consent was approved to obtain from participants during telephone follow-up by the Medical Ethics Committee. Telephone questionnaire in both Chinese and English was presented in supporting information (S1 Appendix).

### Data collection

Demographic characteristics, comorbidities, clinical manifestations, treatment, laboratory findings were extracted from electronic medical records. The records were first accessed on February 15 and last accessed on May 31. Sequential organ failure assessment (SOFA) score,

and laboratory data including white blood cell count, neutrophilic granulocyte percentage, lymphocyte percentage, creatinine, blood urea, aspartate aminotransferase, alanine amino-transferase, total bilirubin, procalcitonin, prothrombin time, activated partial thromboplastin time, hypersensitive troponin I, D-dimer, lactate dehydrogenase were collected on the 1st, 3rd and 7th day after transferred to ICU. Platelet count, six-point ordinal scale scores and oxygen-ation indexes were recorded on the 1st, 3rd, 7th and 10th day. Acute Physiology and Chronic Health Evaluation II (APACHE II) Score was assessed within 24 h after transferred to ICU. The detection of SARS-Cov-2 was conducted by real-time reverse transcriptase polymerase chain reaction (RT-PCR) method or next-generation sequencing. The duration of invasive ventilation, vasopressor days, ICU length of stay, hospital length of stay, and death were also recorded. The primary end point of 28-day 90-day and 180-day mortality were obtained from medical records or telephone follow-up.

## Definition

Six-point ordinal scale was used to evaluate the degree of lung impairment [15]. It was defined as follows: death = 6; ICU admission with extracorporeal membrane oxygenation or mechani-cal ventilation = 5; ICU admission with high-flow oxygen therapy or noninvasive ventila-tion = 4; ICU admission with oxygen therapy = 3; ICU admission but not requiring oxygen therapy = 2; discharged from ICU = 1. According to the test project manual of Leishenshan Hospital and Zhongnan Hospital, the normal range of platelet count was $125–350 \times 10^9$/L. Thus, we defined thrombocytopenia as blood platelet count $\leq 125 \times 10^9$/L and non-thrombocy-topenia as blood platelet count $> 125 \times 10^9$/L. Secondary infections were diagnosed if positive result of a new pathogen was found in blood, sputum or urine cultures. Septic shock was defined bases on Sepsis 3.0 [16]. Acute respiratory distress syndrome was diagnosed based on the Berlin Definition [17]. Acute kidney injury was determined based on KDIGO clinical prac-tice guideline [18]. Acute cardiac injury was diagnosed if the hypersensitive troponin I exceeded the upper limit of reference range ($> 0.04$ ng/mL). And coagulation disorder was diagnosed if PT $> 14$ s or APTT $> 40$ s.

## Statistical analysis

Continuous variables were expressed in the way of mean (standard deviation) or median (interquartile range) according to its distribution type, and categorical variables were pre-sented as numbers and percentages (%). Independent-samples T test would be used when the continuous variables were normally distributed; otherwise, the Mann-Whitney U test would be used. The chi-square test and Fisher's exact test were used to analyze categorical variables. Univariable and multivariable cox regression were used to explore the risk factors related to 28-day and 180-day death. Kaplan-Meier analysis was also used in statistical analysis.

P value less than 0.05 was set as statistically significant. SPSS software version 25.0 (IBM, Armonk, NY) and GraphPad Prism version 8.00 (GraphPad Software Inc., San Diego, Califor-nia) were used in data processing and statistical analysis.

## Results

A total of 167 adult COVID-19 patients (109 from Leishenshan Hospital and 58 from Zhong-nan Hospital) were finally included in this study. 58 patients died during 28 days and a total of 87 patients died within 180 days. All patients ranged in age from 29 to 93, with a median age of 66.00 (IQR, 58.00–77.00), and 67.07% of them were male. Hypertension turned out to be the most common comorbidity, followed by diabetes, coronary heart disease, and cerebrovascular disease. No significant differences in age, gender and comorbidities between two groups were

found. While it was noticed that patients in thrombocytopenia group had significant lower white blood count, platelet count and lactate dehydrogenase, as well as significant higher blood urea, total bilirubin, and SOFA score, compared with non-thrombocytopenia group. Patients with thrombocytopenia had higher creatinine, procalcitonin and high-sensitive cardiac troponin I than the other group though the differences not reach the statistical significance. The clinical characteristics of patients grouped by thrombocytopenia are presented in Table 1.

In univariable analysis, age, white blood count, neutrophil granulocyte percentage, lymphocyte percentage, thrombocytopenia, total bilirubin, prothrombin time, activated partial prothrombin time, D-dimer, and lactate dehydrogenase at ICU admission were related to 28-day mortality. These above variables were all included in multivariable cox regression analysis, and older age, lower platelet count and longer active partial thromboplastin time were finally proved to be relative risk factors of 28-day death. In the same way, we found that older age, higher white blood cell, lower platelet count, and longer activated partial thromboplastin time were risk factors of 180-day death (Table 2).

To better understanding the association between thrombocytopenia and organ functions, the changes of laboratory data and rating scores over time in patients with and without thrombocytopenia are presented. It showed that six-point ordinal scale score and SOFA score as well as oxygenation index had tendencies to deteriorate over time in patients with thrombocytopenia. Other significant differences were also noticed on blood urea, aspartate aminotransferase, lymphocyte percentage, procalcitonin, and white blood count with time (Table 3)

The patients with lower platelet count had worse respiratory function at admission (S1 Fig). The time curves of platelet count, six-point ordinal scale score and oxygenation index were also drawn. For non-survival group, the platelet and oxygenation index gradually decreased over time while the six-point ordinal scale score showed an upward trend, which meant the worse respiratory function was probably related to the lower platelets (Fig 1A and 1B). But the downtrends were not seen in survival group (Fig 1C and 1D), which may come down to the different incidences of thrombocytopenia and severity of disease (S1 Table). Furthermore, Kaplan-Meier and Log Rank analysis showed that patients with thrombocytopenia at ICU admission had significant lower probability of survival than the other group (Fig 2). The median survival time in patients with thrombocytopenia was 17 (IQR, 7.5–180) days.

As for clinical outcomes, the significant differences on 28-day mortality (56.10% vs 27.78%, p<0.01), 90-day mortality (65.85% vs 45.24%, p = 0.02), and 180-day mortality (65.85% vs 47.62%, p = 0.04) between patients with and without thrombocytopenia were demonstrated. In addition, patients with thrombocytopenia had significant higher proportion of receiving antiviral therapy (60.98% vs 42.86%, p = 0.04), and invasive ventilation (58.54% vs 38.89%, p = 0.03), but shorter ICU length of stay (median day, 10.50 [IQR, 5.50–15.80] vs 13.00 [IQR, 8.00–23.00], p = 0.03), which might be related to more early death events (S2 Fig). It appeared that patients with thrombocytopenia were much more in need for vasopressor therapy. Moreover, patients with thrombocytopenia were much more likely to develop septic shock, arrhythmia, coagulation disorder, compared with patients without thrombocytopenia. Most deaths (96.55%) occurred within 90 days, suggesting that COVID-19 was as an acute blow to human health. No significant differences in long-term prognosis between groups were found, but patients with thrombocytopenia had higher proportion of extra oxygen supply (35.71% vs 13.64%, p = 0.06) (Table 4).

## Discussion

This is an observational study of the 180-day prognosis and clinical development in critically ill COVID-19 patients. Our study did a long-term follow up and confirmed a few risk factors

**Table 1. Baseline characteristics of the study patients infected with SARS-CoV-2.**

| Variables | Total (n = 167) | Thrombocytopenia (n = 41) | Non-thrombocytopenia (n = 126) | P value |
|---|---|---|---|---|
| Age, years | 66.00 [58.00–77.00] | 68.00 [61.50–78.50] | 66.00 [57.00–77.00] | 0.16 |
| Gender | | | | 0.34 |
| Male | 112 (67.07%) | 25 (60.98%) | 87 (69.05%) | |
| Female | 55 (32.93%) | 16 (39.02%) | 39 (30.95%) | |
| **Comorbidities** | | | | |
| Hypertension | 85 (50.90%) | 22 (53.66%) | 63 (50.00%) | 0.68 |
| Diabetes | 48 (28.74%) | 10 (24.39%) | 38 (30.16%) | 0.48 |
| Coronary heart disease | 39 (23.35%) | 7 (17.07%) | 32 (25.40%) | 0.27 |
| Chronic lung disease | 12 (7.19%) | 3 (7.32%) | 9 (7.14%) | 0.97 |
| Cerebrovascular disease | 36 (21.56%) | 7 (17.07%) | 29 (23.02%) | 0.42 |
| Chronic renal failure | 19 (11.38%) | 4 (9.76%) | 15 (11.90%) | 1.00 |
| Malignant tumor | 9 (5.39%) | 3 (7.32%) | 6 (4.76%) | 0.69 |
| **Vital signs** | | | | |
| Heart rate, beats per minute | 92.25 (19.66) | 93.15 (21.21) | 91.95 (19.21) | 0.74 |
| Respiratory rate, breaths per minute | 23.61 (9.79) | 23.41 (7.13). | 23.67 (10.54) | 0.86 |
| Mean arterial pressure, mmHg | 91.35 (14.82) | 93.78 (14.81) | 90.56 (14.80) | 0.23 |
| **Laboratory findings** | | | | |
| WBC, ×109/L | 8.13 [6.14–11.85] | 7.17 [4.68–9.89] | 8.75 [6.44–12.26] | <0.01 |
| Neutrophilic granulocyte percentage, % | 84.17 [75.05–91.18] | 87.50 [71.70–91.60] | 83.40 [75.10–90.95] | 0.61 |
| Lymphocyte percentage, % | 8.25 [4.10–14.53] | 7.20 [4.00–14.60] | 8.50 [4.25–14.55] | 0.77 |
| Platelet count, ×109/L | 190.00 [126.00–236.00] | 108.00 [77.00–119.50] | 211.50 [176.00–269.50] | <0.01 |
| Creatinine, μmol/L | 68.45 [55.38–107.85] | 78.80 [62.30–126.45] | 65.40 [53.55–97.35] | 0.06 |
| Blood urea, mmol/L | 10.05 [6.28–15.78] | 11.50 [8.80–16.07] | 9.00 [5.75–15.70] | 0.04 |
| AST, IU/L | 36.00 [22.00–63.00] | 50.00 [26.00–80.00] | 36.00 [21.25–61.50] | 0.13 |
| ALT, IU/L | 31.00 [19.60–47.50] | 34.00 [20.00–43.50] | 29.00 [19.40–48.75] | 0.46 |
| Total bilirubin, μmol/L | 10.50 [7.45–15.12] | 13.30 [8.05–22.80] | 10.20 [7.43–13.95] | 0.02 |
| Procalcitonin, ng/mL | 0.17 [0.07–0.57] | 0.29 [0.10–0.72] | 0.15 [0.07–0.47] | 0.06 |
| PT, seconds | 12.95 [11.90–14.45] | 13.20 [11.80–14.85] | 12.90 [11.95–14.20] | 0.96 |
| APTT, seconds | 31.60 [27.88–36.60] | 31.10 [26.75–39.75] | 31.60 [27.95–36.35] | 0.92 |
| hs-CTn I, ng/mL | 0.02 [0.01–0.06] | 0.04 [0.01–0.12] | 0.02 [0.01–0.04] | 0.09 |
| D-dimer, μg/L | 1.99 [0.90–6.61] | 2.26 [0.44–21.22] | 1.97 [0.90–4.42] | 0.22 |
| LDH, U/L | 354.00 [251.00–554.00] | 504.00 [303.50–623.00] | 328.00 [248.00–508.25] | <0.01 |
| APACHE II | 12.00 [9.00–16.00] | 12.00 [10.50–15.50] | 12.00 [9.00–16.00] | 0.72 |
| SOFA | 4.00 [3.00–7.00] | 5.00 [4.00–7.50] | 4.00 [2.00–5.00] | <0.01 |
| Six-point ordinal scale on day 1 | 4.00 [3.00–5.00] | 4.00 [3.00–5.00] | 4.00 [3.00–5.00] | 0.31 |
| 2-hospital admission, not requiring supplemental oxygen | 2(1.20%) | 2(4.88%) | 0.00 | |
| 3-hospital admission, requiring supplemental oxygen | 60 (35.93%) | 9 (21.95%) | 51 (40.48%) | |
| 4-hospital admission, requiring high-flow oxygen therapy or non-invasive ventilation | 60 (35.93%) | 18 (43.90%) | 42 (33.33%) | |
| 5-hospital admission, requiring invasive ventilation or ECMO | 45 (26.95%) | 12 (29.27%) | 33 (26.19%) | |

(WBC) White blood cell; (AST) Aspartate aminotransferase; (ALT) Alanine aminotransferase; (PT) Prothrombin time; (APTT) Activated partial thromboplastin time; (hs-CTn I) Hypersensitive cardiac troponin I; (LDH) Lactate dehydrogenase; (APACHE II) Acute Physiology and Chronic Health Status Score II; (SOFA) Sequential Organ Failure Assessment.

**Table 2. Risk factors associated with 28-day mortality and 180-day mortality.**

| Variables | 28-day mortality | | | | 180-day mortality | | | |
|---|---|---|---|---|---|---|---|---|
| | Univariable RR [95% CI] | p value | Multivariable RR [95% CI] | p value | Univariable RR [95% CI] | p value | Multivariable RR [95% CI] | p value |
| **Demographic and clinical characteristics** | | | | | | | | |
| Age, years | 1.02 [1.00–1.04] | 0.03 | 1.03 [1.00–1.05] | 0.04 | 1.03 [1.01–1.05] | <0.01 | 1.04 [1.02–1.06] | <0.01 |
| Gender | 0.88 [0.51–1.51] | 0.64 | | | 0.94 [0.60–1.47] | 0.80 | | |
| Diabetes | 1.00 [0.57–1.77] | 0.99 | | | 0.93 [0.58–1.49] | 0.77 | | |
| Hypertension | 1.51 [0.89–2.54] | 0.13 | | | 1.41 [0.92–2.16] | 0.11 | | |
| Coronary heart disease | 1.35 [0.76–2.41] | 0.31 | | | 1.51 [0.96–2.40] | 0.07 | | |
| Chronic lung disease | 0.64 [0.20–2.04] | 0.45 | | | 0.82 [0.36–1.87] | 0.63 | | |
| Chronic renal failure | 1.41 [0.67–2.98] | 0.37 | | | 1.42 [0.77–2.61] | 0.26 | | |
| Malignant tumor | 1.26 [0.46–3.48] | 0.66 | | | 1.51 [0.66–3.46] | 0.33 | | |
| Cerebrovascular disease | 0.91 [0.48–1.71] | 0.76 | | | 1.10 [0.67–1.79] | 0.71 | | |
| **Laboratory findings** | | | | | | | | |
| WBC, ×10⁹/L | 1.06 [1.02–1.10] | <0.01 | 1.04 [0.98–1.11] | 0.16 | 1.07 [1.04–1.10] | <0.01 | 1.06 [1.01–1.12] | 0.03 |
| Neutrophilic granulocyte percentage, % | 1.05 [1.02–1.08] | <0.01 | 1.01 [0.91–1.12] | 0.84 | 1.05 [1.02–1.07] | <0.01 | 0.95 [1.03–1.12] | 0.44 |
| Lymphocyte percentage, % | 0.93 [0.89–0.98] | <0.01 | 0.96 [0.84–1.10] | 0.55 | 0.95 [0.91–0.98] | <0.01 | 1.00 [0.90–1.11] | 0.99 |
| Platelet count, ×10⁹/L | | | | | | | | |
| Thrombocytopenia, ≤125 | 2.79 [1.65–4.74] | <0.01 | 2.98 [1.48–6.02] | <0.01 | 2.08 [1.32–3.29] | <0.01 | 2.45 [1.32–4.57] | <0.01 |
| Non-thrombocytopenia, >125 | 1 (ref) | - | - | - | 1 (ref) | - | - | - |
| Creatinine, ×10⁹/L | 1.00 [1.00–1.00] | 0.41 | | | 1.00 [1.00–1.00] | 0.22 | | |
| Blood urea, mmol/L | 1.00 [0.99–1.01] | 0.60 | | | 1.00 [1.00–1.01] | 0.60 | | |
| AST, IU/L | 1.00 [1.00–1.00] | 0.40 | | | 1.00 [1.00–1.00] | 0.88 | | |
| ALT, IU/L | 1.00 [1.00–1.00] | 0.84 | | | 1.00 [1.00–1.00] | 0.65 | | |
| Total bilirubin, μmol/L | 1.03 [1.00–1.05] | 0.05 | 0.99 [0.96–1.03] | 0.70 | 1.03 [1.01–1.05] | 0.01 | 0.99 [0.97–1.02] | 0.68 |
| Procalcitonin, ng/mL | 1.00 [0.98–1.01] | 0.64 | | | 0.99 [0.98–1.01] | 0.48 | | |
| PT, seconds | 1.11 [1.04–1.18] | <0.01 | 0.98 [0.87–1.10] | 0.73 | 1.13 [1.06–1.19] | <0.01 | 1.00 [0.90–1.10] | 0.93 |
| APTT, seconds | 1.02 [1.00–1.04] | 0.02 | 1.02 [1.00–1.04] | 0.05 | 1.03 [1.01–1.04] | <0.01 | 1.00 [1.02–1.04] | 0.01 |
| hs-CTn I, ng/mL | 1.05 [0.80–1.37] | 0.75 | | | 1.04 [0.83–1.29] | 0.76 | | |
| D-dimer, μg/L | 1.02 [1.01–1.03] | <0.01 | 0.99 [0.97–1.01] | 0.50 | 1.02 [1.01–1.03] | <0.01 | 0.98 [1.00–1.01] | 0.57 |
| LDH, U/L | 1.00 [1.00–1.00] | 0.04 | 1.00 [1.00–1.00] | 0.35 | 1.00 [1.00–1.00] | 0.01 | 1.00 [1.00–1.00] | 0.15 |

(RR) relative risk; (CI) confidence Interval; (WBC) White blood cell; (AST) Aspartate aminotransferase; (ALT) Alanine aminotransferase; (PT) Prothrombin time; (APTT) Activated partial thromboplastin time; (Hs-CTn I) hypersensitive troponin I; (LDH) Lactate dehydrogenase.

of 180-day mortality in critically ill COVID-19 patients. Besides, our study found that the decrease of platelet was associated with poor respiratory function, which were further proved by the time curves of six-point ordinal scale, oxygenation index and platelet count. Our study also confirmed that thrombocytopenia was associated with mortality in critically ill COVID-19 patients. 86.6% of death events occurred within 90 days, and 66.67% of them occurred within 28 days, which means this disease developed rapidly and led to death in a short time in high-risk patients.

## Thrombocytopenia and respiratory function

As a vital blood component, platelets play a crucial role in initial clot or thrombus formation. Other than that, platelets are believed to be the first responders in innate immune and can interact with pathogens including bacteria and virus through multiple platelet surface

**Table 3. Subsequent changes of rating scores and laboratory findings.**

| Variables | Days | Thrombocytopenia | | Non-thrombocytopenia | | P value |
|---|---|---|---|---|---|---|
| | | Mean/Median | SD/IQR | Mean/Median | SD/IQR | |
| six-point ordinal scale | 1 | 4.00 | 3.00–5.00 | 4.00 | 3.00–5.00 | 0.31 |
| | 3 | 5.00 | 4.00–5.00 | 4.00 | 3.00–5.00 | <0.01 |
| | 7 | 5.00 | 3.50–5.00 | 4.00 | 3.00–5.00 | 0.01 |
| Oxygenation index | 1 | 113.20 | 81.64–179.00 | 135.75 | 83.38–238.14 | 0.11 |
| | 3 | 144.51 | 83.15–202.73 | 185.71 | 126.03–261.92 | 0.01 |
| | 7 | 130.67 | 82.80–212.00 | 174.50 | 107.72–269.69 | 0.05 |
| SOFA | 1 | 5.00 | 4.00–7.50 | 4.00 | 2.00–5.00 | <0.01 |
| | 3 | 6.00 | 4.00–7.00 | 4.00 | 3.00–6.00 | <0.01 |
| | 7 | 6.00 | 3.00–9.00 | 3.00 | 2.00–6.00 | <0.01 |
| Creatinine, μmol/L | 1 | 141.26 | 208.15 | 129.64 | 193.36 | 0.06 |
| | 3 | 130.90 | 192.95 | 115.74 | 156.59 | 0.26 |
| | 7 | 145.77 | 171.53 | 123.83 | 171.21 | 0.06 |
| blood urea, mmol/L | 1 | 15.08 | 8.24 | 13.44 | 22.43 | 0.01 |
| | 3 | 11.77 | 8.54 | 9.00 | 6.02 | 0.03 |
| | 7 | 13.66 | 9.98 | 10.03 | 7.24 | 0.12 |
| AST, IU/L | 1 | 158.90 | 647.20 | 97.21 | 402.82 | 0.13 |
| | 3 | 72.23 | 116.50 | 69.71 | 188.23 | 0.02 |
| | 7 | 77.96 | 121.33 | 93.73 | 478.35 | 0.02 |
| ALT, IU/L | 1 | 77.20 | 236.77 | 71.26 | 228.68 | 0.46 |
| | 3 | 49.55 | 43.04 | 67.91 | 164.94 | 0.20 |
| | 7 | 50.54 | 43.04 | 67.91 | 164.94 | 0.19 |
| Lymphocyte percentage, % | 1 | 10.04 | 7.81 | 10.31 | 7.82 | 0.77 |
| | 3 | 8.32 | 4.58 | 11.74 | 8.01 | 0.10 |
| | 7 | 7.88 | 6.33 | 13.11 | 9.31 | 0.03 |
| Procalcitonin, mg/mL | 1 | 0.76 | 1.11 | 4.00 | 24..98 | 0.05 |
| | 3 | 8.10 | 31.29 | 2.19 | 10.64 | 0.01 |
| | 7 | 3.37 | 9.02 | 3.28 | 24.70 | 0.04 |
| PT, seconds | 1 | 14.24 | 4.49 | 13.42 | 2.09 | 0.96 |
| | 3 | 14.13 | 2.21 | 13.83 | 2.70 | 0.25 |
| | 7 | 14.20 | 3.58 | 13.44 | 2.36 | 0.48 |
| APTT, seconds | 1 | 34.62 | 10.08 | 33.40 | 10.54 | 0.56 |
| | 3 | 36.63 | 12.79 | 34.89 | 13.58 | 0.63 |
| | 7 | 35.87 | 12.14 | 34.47 | 8.54 | 0.91 |
| WBC, ×10$^9$/L | 1 | 7.36 | 3.44 | 10.24 | 6.00 | 0.01 |
| | 3 | 8.04 | 4.24 | 9.62 | 5.87 | 0.15 |
| | 7 | 10.11 | 5.13 | 9.76 | 6.36 | 0.50 |
| hs-CTn I, ng/mL | 1 | 0.46 | 1.70 | 0.17 | 0.78 | 0.07 |
| | 3 | 0.36 | 0.78 | 0.16 | 0.38 | 0.48 |
| | 7 | 0.38 | 0.77 | 0.15 | 0.42 | 0.22 |

(SD) Standard deviation; (IQR) Inter quartile range; (SOFA) Sequential Organ Failure Assessment; (AST) Aspartate aminotransferase; (ALT) Alanine aminotransferase; (PT) Prothrombin time; (APTT) Activated partial thromboplastin time; (WBC) White blood count; (hs-CTn I) hypersensitive troponin I.

receptors [19–21]. In addition to against microbial insults directly, platelets aid primary immune cells in the clearance of pathogens. Platelets can secret cytokines such as IL-1β, which will lead to endothelium high permeability, as well as recruitment and attachment of

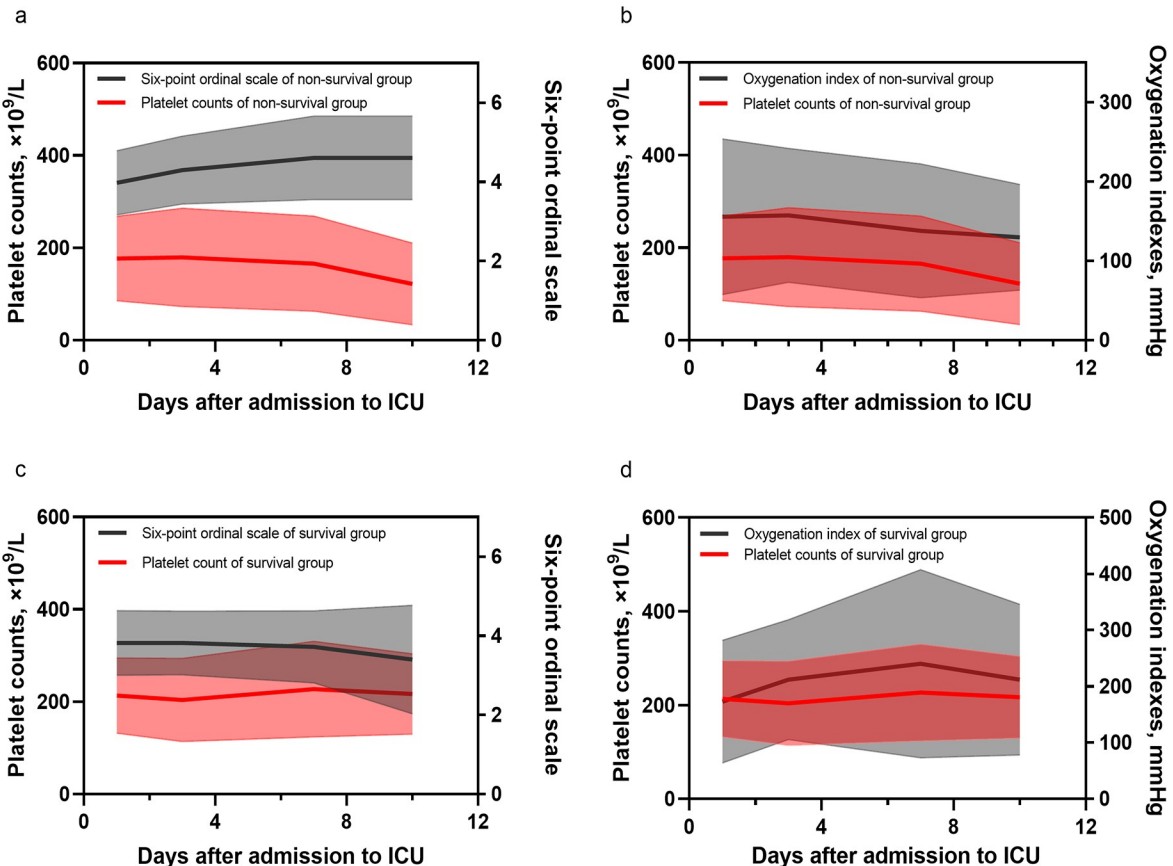

**Fig 1. Time curves of platelet count, six-point ordinal scale score and oxygenation index.** Shaded areas show the standard error while the curves show the means of platelet, six-point ordinal scale and oxygenation index. a. Time curve of platelet count and oxygenation index of non-survival group. b. Time curve of platelet count and six-point ordinal scale of non-survival group. c. Time curve of platelet count and oxygenation index of survival group. d. Time curve of platelet count and six-point ordinal scale of survival group.

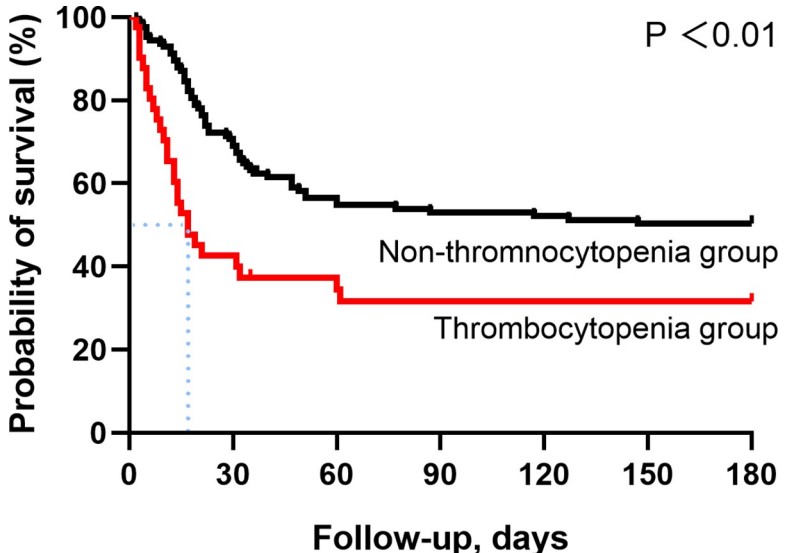

**Fig 2. Survival curve in patients with thrombocytopenia and non-thrombocytopenia.**

**Table 4. Complications, treatments and clinical outcomes.**

| Variables | Total (n = 167) | Thrombocytopenia (n = 41) | Non-thrombocytopenia (n = 126) | P value |
|---|---|---|---|---|
| **Treatments** | | | | |
| **Antiviral therapy** | 79 (47.31%) | 25 (60.98%) | 54 (42.86%) | 0.04 |
| Arbidol | 28 (16.77%) | 12 (29.27%) | 16 (12.70%) | 0.01 |
| Ribavirin | 19 (11.38%) | 6 (14.63%) | 13 (10.32%) | 0.57 |
| Entecavir | 1 (0.60%) | 1 (2.44%) | 0 | 0.25 |
| Interferon | 21 (12.57%) | 3 (7.32%) | 18 (14.29%) | 0.29 |
| Chloroquine phosphate | 3 (1.80%) | 1 (2.44%) | 2 (1.59%) | 0.57 |
| **Antibiotic therapy** | 154 (92.22%) | 38 (92.68%) | 116 (92.06%) | 0.90 |
| Imipenem/meropenem | 73 (43.71%) | 23 (31.5%) | 50 (39.68%) | 0.07 |
| Quinolone | 106 (63.47%) | 27 (65.85%) | 79 (62.70%) | 0.72 |
| Cephalosporin | 95 (56.89%) | 24 (58.54%) | 71 (56.35%) | 0.81 |
| Piperacillin sodium and sulbactam sodium | 23 (13.77%) | 3 (7.32%) | 20 (15.87%) | 0.17 |
| Azithromycin | 5 (2.99%) | 1 (2.44%) | 4 (3.17%) | 0.81 |
| Linezolid | 31 (18.56%) | 5 (12.20%) | 26 (20.63%) | 0.23 |
| Polymyxin B | 12 (7.19%) | 3 (7.32%) | 9 (7.14%) | 0.97 |
| Vancomycin | 20 (11.98%) | 6 (14.63%) | 14 (11.11%) | 0.55 |
| Daptomycin | 1 (0.60%) | 0 | 1 (0.79%) | 1.00 |
| Tigecycline | 17 (10.18%) | 4 (9.76%) | 13 (10.32%) | 1.00 |
| **Antifungal therapy** | 34 (20.36%) | 9 (21.95%) | 25 (19.84%) | 0.79 |
| Caspofungin | 12 (7.19%) | 4 (9.76%) | 8 (6.35%) | 0.46 |
| Fluconazole | 6 (3.59%) | 1 (2.44%) | 5 (3.97%) | 1.00 |
| Voriconazole | 13 (7.78%) | 3 (7.32%) | 10 (7.94%) | 1.00 |
| **Immunomodulator** | | | | |
| Thymalfasin/ Thymosin | 21 (12.57%) | 4 (9.76%) | 17 (13.49%) | 0.53 |
| XUE BI JING injection | 20 (11.98%) | 8 (19.51%) | 12 (9.52%) | 0.09 |
| Ulinastatin | 6 (3.59%) | 2 (4.88%) | 4 (3.17%) | 0.64 |
| Corticosteroids therapy | 73 (43.71%) | 21 (51.22%) | 52 (41.27%) | 0.27 |
| **Respiratory support** | | | | |
| High-flow oxygen therapy | 63 (37.72%) | 14 (34.15%) | 49 (38.89%) | 0.59 |
| Non-invasive ventilation | 59 (35.33%) | 16 (39.02%) | 43 (34.13%) | 0.57 |
| Invasive ventilation | 75 (44.91%) | 24 (58.54%) | 51 (40.48%) | 0.04 |
| Duration of invasive ventilation, days | 11.00 [6.00–20.00] | 12.00 [7.00–22.00] | 8.50 [5.00–13.00] | 0.08 |
| Vasopressors | 70 (41.92%) | 22 (53.66%) | 48 (38.10%) | 0.08 |
| CRRT | 34 (20.40%) | 8 (19.50%) | 26 (20.60%) | 0.88 |
| ECMO | 21 (12.60%) | 4 (9.80%) | 17 (13.50%) | 0.53 |
| **Complications** | | | | |
| Secondary infections | 98 (57.68%) | 21 (51.22%) | 77 (61.11%) | 0.26 |
| Septic shock | 48 (28.74%) | 17 (41.46%) | 31 (24.60%) | 0.04 |
| ARDS | 148 (88.62%) | 38 (92.68%) | 110 (87.30%) | 0.35 |
| Acute kidney injury | 34 (20.48%) | 11 (26.83%) | 23 (18.40%) | 0.25 |
| Arrhythmia | 35 (20.96%) | 17 (41.46%) | 18 (14.29%) | <0.01 |
| Acute cardiac injury | 57 (34.13%) | 19 (46.34%) | 38 (30.16%) | 0.06 |
| Coagulation disorder | 43 (25.75%) | 18 (43.90%) | 25 (19.84%) | <0.01 |
| **Main outcomes** | | | | |
| ICU length of stay, days | 12.00 [7.00–21.00] | 10.50 [5.50–15.75] | 13.00 [8.00–23.00] | 0.03 |
| Hospital length of stay, days | 21.00 [15.00–35.00] | 20.00 [11.00–26.50] | 23.00 [15.75–36.25] | 0.06 |
| 28-day mortality | 58 (34.73%) | 23 (56.10%) | 35 (27.78%) | <0.01 |

*(Continued)*

**Table 4.** (Continued)

| Variables | Total (n = 167) | Thrombocytopenia (n = 41) | Non-thrombocytopenia (n = 126) | P value |
|---|---|---|---|---|
| 90-day mortality | 84 (50.29%) | 27 (65.85%) | 57 (45.24%) | 0.02 |
| 180-day mortality | 87 (52.1%) | 27 (65.85%) | 60 (47.62%) | 0.04 |
| **Survival prognosis in 180 days** | | | | |
| Survival, numbers | 80 | 14 | 66 | |
| Extra oxygen supply | 14 (17.50%) | 5 (35.71%) | 9 (13.64%) | 0.06 |
| Exertional dyspnea | 17 (21.25%) | 3 (21.43%) | 14 (21.21%) | 1.00 |
| weakness | 15 (18.75%) | 3 (21.43%) | 12 (18.18%)) | 0.72 |

(SD) standard deviation; (IQR) Inter quartile range; (SOFA) Sequential Organ Failure Assessment; (CRRT) continuous renal replacement therapy; (ECMO), extracorporeal membrane oxygenation; (ARDS) acute respiratory distress syndrome.

leukocytes to the endothelium [22, 23]. Platelets can also interact with other immune cells such as neutrophils, monocytes, dendritic cells, and lymphocytes [23–25].

The decrease of platelet was associated with poor respiratory function in critically ill patients. Platelets directly interact with viral pathogens through the pathogen recognition receptors such as protease-activated receptor 4 (PAR4) and glycoprotein IIIa (GPIIIa), and this interaction can lead to platelet activation, which is associated with lung inflammation as well as the severity of viral infections, lung injury and death [26, 27]. The interactions between platelets, leukocytes and endothelial cells are also critical in the pathogenesis of acute lung injury [28, 29]. All of these evidences indicated that platelets played a role in lung injury caused by SARS-CoV-2. In a phase IIb case control study [30], antiplatelet therapy improved ventilation/perfusion in COVID-19 patients with severe respiratory failure, which further supported the point that the platelet participated in the progression of lung injury and respiratory failure in critically ill COVID-19 patients. Endothelial injury was reported to be directly caused by SARS-CoV-2 [31]. The endothelial injury and cytokines released from endothelial cells can also lead to platelet activation and aggregation, leading to microthrombi formation and thrombocytopenia and lung injury [31, 32]. Thus, the association between thrombocytopenia and respiratory dysfunction is quite intricate and cannot be explained by simple causation.

## Mechanism of thrombocytopenia

Thrombocytopenia is one of the most common laboratory abnormalities in critically ill patients [33–35]. The mechanism of thrombocytopenia in COVID-19 patients might be related to decreased production, increased consumption and destruction of platelets [36].

First, SARS-CoV-2 infections may reduce the production of platelet. Coronavirus can infect and inhibit the growth of hematopoietic stem progenitor cells and the megakaryocytes [37]. SARS-CoV-2 may also decrease the production of platelets by entering the bone marrow cells and platelet cells through CD13 and CD66a [38–41]. Besides, evidence [42] showed that amounts of megakaryocytes release platelet when they circulate through the lungs. Pulmonary capillary injury and microthrombi formation can block the circulation and platelet release in COVID-19 patients [31]. Second, SARS-CoV-2 infections may lead to increased consumption of platelet. COVID-19 patients often have the extremity deep vein thrombosis and alveolar capillary microthrombi [31, 43, 44]. Platelet play a role in innate immune and can be activated by bacteria and virus [19–21]. Influenza A H1N1 and EMCV can trigger platelet activation in FcγRIIa-dependent and Toll-like receptor 7-dependent way separately [45, 46]. As a member of single-stranded RNA virus just like influenza A H1N1 and EMCV, SARS-CoV-2 may also trigger platelet activation and aggregation, leading to platelet consumption [47]. Besides,

SARS-CoV-2 virus was found within endothelial cells, suggesting that the endothelial injury may be caused by direct viral effects [31]. As a result of endothelial injury, sub-endothelial collagen exposure promotes innate adhesion of platelets to collagen through von Willebrand factor, which is vital in the adhesion process and significantly increased in COVID-19 patients [32, 48]. Meanwhile, activated endothelial cells will facilitate the expression of tissue factor (TF), and may activated coagulation cascade culmination [32]. Third, it's reported that SARS-CoV-2 might specifically destroy platelets through autoantibodies [36]. Antiplatelet antibodies were also confirmed to be the cause of vancomycin-induced thrombocytopenia [49, 50]. Other than that, Drugs such as linezolid, ribavirin, interferon, cephalosporin and chloroquine phosphate may result in drug-associated thrombocytopenia [50–53]. ECMO and CRRT have also been reported to be associated with decrease of platelet counts [54–57]. High incidences of secondary infections and septic shock were seen in our study, and the intense inflammation response will promote thrombosis and lead to thrombocytopenia as well [33, 58].

## Mortality risk factors

Older age, lower platelet count, higher white blood cell has been proven to relate with clinical worsening and poor outcome in many studies [10, 11, 59, 60]. Longer activated partial thromboplastin time was one of the characteristics in blood system of COVID-19 patients. Previous study reported D-dimer, prothrombin time, and activated partial thromboplastin time could be used as indicators for disease prognosis [61]. We also found longer activated partial thromboplastin time was a risk factor of death, though D-dimer and prothrombin time didn't show significance in our study.

## Thrombocytopenia and mortality

The lower platelet count has been reported to be a marker of poor prognosis, not only in COVID-19 patients but also in different population of critically ill patients [10–12, 33]. Thrombocytopenia has been demonstrated to be associated with increased length of ICU and hospital stay as well as higher mortality [33]. Our findings were consistent with previous studies, and demonstrated that thrombocytopenia had almost threefold risk of death in 28 days and more than double risk of death in 180 days than those with non-thrombocytopenia at admission. However, the intrinsic connection between thrombocytopenia and death was not clear yet. Respiratory failure, followed by multiple septic shock and organ failure, turned out to be the main cause of death of COVID-19 patients [62]. As the thrombocytopenia induced the development of organ impairment including renal failure, acute lung injury, respiratory distress syndrome, vascular leakage syndromes and septic shock [33, 63], thrombocytopenia may relate to death in a roundabout way.

## Limitations

This study has several limitations. First, this is an observational study, and it is hard to exclude all potential confounders, and keep data integrity and consistency, so the role of some variables might be underestimated in predicting the prognosis. Second, our study proved that low platelets were associated with subsequent poor respiratory function and mortality, but it is difficult to confirm if the association is causality. It needs more basic researches, animal experiments or prospective trials to confirm that. Third, we didn't measure the platelets count and coagulation functions in 180 day-survivors. Nevertheless, this observational study truly provided some evidence on the association between thrombocytopenia and progress of organ functions, and long-term outcome in critically ill COVID-19.

## Conclusion

Thrombocytopenia at admission was associated with mortality of COVID-19. The subsequent change of platelet was associated with respiratory function as well as other organ functions, and finally contributed to the death in patients with severe COVID-19 in ICU.

## Supporting information

**S1 Appendix. Telephone questionnaire.**
(PDF)

**S2 Appendix. Dataset.**
(XLSX)

**S1 Fig. The platelet counts and oxygenation indexes at admission.**
(PDF)

**S2 Fig. 14-day Survival Curve in Patients with and without Thrombocytopenia.**
(PDF)

**S1 Table. Subsequent changes of organ function.**
(PDF)

## Acknowledgments

We acknowledge all health-care workers involved in the diagnosis and treatment of patients in Wuhan.

## Author Contributions

**Conceptualization:** Yuan Zhu, Jing Zhang, Yiming Li, Fang Liu, Qing Zhou, Zhiyong Peng.

**Data curation:** Yuan Zhu, Jing Zhang, Yiming Li, Qing Zhou, Zhiyong Peng.

**Formal analysis:** Yuan Zhu, Jing Zhang.

**Investigation:** Yuan Zhu, Jing Zhang, Fang Liu.

**Methodology:** Yuan Zhu, Jing Zhang, Yiming Li, Zhiyong Peng.

**Project administration:** Qing Zhou, Zhiyong Peng.

**Resources:** Yuan Zhu.

**Supervision:** Yuan Zhu, Jing Zhang, Yiming Li, Fang Liu, Qing Zhou, Zhiyong Peng.

**Writing – original draft:** Yuan Zhu, Jing Zhang.

**Writing – review & editing:** Yuan Zhu, Jing Zhang, Yiming Li, Fang Liu, Qing Zhou, Zhiyong Peng.

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
