## [Decision Letter · Decision Letter 0]

1 Dec 2020

PONE-D-20-31221

Association between thrombocytopenia and 180-day prognosis of COVID-19 patients in intensive care units: a two-center observational study.

PLOS ONE

Dear Dr. Peng,

Thank you for submitting your manuscript to PLOS ONE. After careful consideration, we feel that it has merit but does not fully meet PLOS ONE’s publication criteria as it currently stands. Therefore, we invite you to submit a revised version of the manuscript that addresses the points raised during the review process.

In revising this manuscript you should focus on the novelty of your findings and how it is different to other papers.

We look forward to receiving your revised manuscript.

Kind regards,

Dermot Cox

Academic Editor

PLOS ONE

Journal Requirements:

2. Please include additional information regarding the telephone questionnaire/interview guide used in the study and ensure that you have provided sufficient details that others could replicate the analyses. For instance, if it is not under a copyright more restrictive than CC-BY, please include a copy, in both the original language and English, as Supporting Information.

3. Please include the date(s) on which you accessed the databases or records to obtain the data used in your study.

4.We note that you have indicated that data from this study are available upon request. PLOS only allows data to be available upon request if there are legal or ethical restrictions on sharing data publicly. For information on unacceptable data access restrictions, please see http://journals.plos.org/plosone/s/data-availability#loc-unacceptable-data-access-restrictions.

Additional Editor Comments (if provided):

You should expand a little on the section on mechanism of thrombocytopenia. For instance providing an example of viruses that are associated with thrombocytopenia with known mechanism of actions. I have attached a review on this topic which should assist in identifying examples (Alonso AL, Cox D. Platelet interactions with viruses and parasites. Platelets. 2015;26(4):317-23). Furthermore, Covid-19 causes a viral sepsis which is very similar to bacterial sepsis and there have been many studies on how bacteria cause thrombocytopenia. Here is a review on this which may help you identify suitable examples (Kerrigan SW, Devine T, Fitzpatrick G, et al. Early Host Interactions That Drive the Dysregulated Response in Sepsis. Frontiers in Immunology. 2019;10(1748)).

Reviewers' comments:

Reviewer's Responses to Questions

**Comments to the Author**

1. Is the manuscript technically sound, and do the data support the conclusions?

Reviewer #1: Yes

Reviewer #2: Yes

2. Has the statistical analysis been performed appropriately and rigorously? 

Reviewer #1: Yes

Reviewer #2: Yes

3. Have the authors made all data underlying the findings in their manuscript fully available?

Reviewer #1: Yes

Reviewer #2: Yes

4. Is the manuscript presented in an intelligible fashion and written in standard English?

Reviewer #1: Yes

Reviewer #2: Yes

5. Review Comments to the Author

Reviewer #1: In this study, the authors try to correlate platelet counts with deterioration of respiratory function and prognosis. Such studies already exist

Major

1. The authors should try and explain how this study is different to papers published already on thrombocytopenia and poor prognosis in COVID-19 (for example; ?longer follow-up)

2. Several factors were noted to be associated with poor prognosis on univariate analysis. How important is platelet count among these

3. The significant finding is the correlation with respiratory function. it would be useful to build upon this finding

4. You explain on page 20, line 162...the worse respiratory function was probably related to the lower platelets. But the tendencies in survival group didn’t seem to make sense -- could you elaborate

5. page 23, line 193 says, shorter ICU length of stay, which might be related to more early death events -- is it possible to confirm this

6. page 27, line 212, Most death events occurred within 90 days, which means the COVID-19 is an acute attack to those affected -- what is meant by acute attack

7. In the mechanism of thrombocytopenia, the role of high Von Willebrand levels is worth explaining

8. the discussion on thrombocytopenia and respiratory function would benefit from some explanation for causation. The description here is about role of platelets rather than thrombocytopenia

9. use mortality risk factors rather than Death risk factors

Reviewer #2: I would like to thank the editor for giving me the opportunity to review this article. The authors reported a detailed data of a cohort of sars-cov2 patients and thoroughly investigated the prognostic value of thrombocytopenia in this group of patients.

Major Comments

1. Results

- One the major findings of the study was that both thrombocytopenia and prolonged aPTT were associated with higher 28-day and 180-day mortality. This finding may suggest that patient who died would have higher incidence of disseminated intravascular coagulation (DIC). Tang N et al reported that the incidence of DIC in patients who died because of sars-cov2 pneumonia was as high as 71.4 % (J Thromb Haemost, 2020). This parameter was not investigated per se by the authors. I would advise the authors to define it and to investigate its value in predicting mortality.

- Results: Table 3 displays the data on day 2, 2 and 3 while the authors mentioned in the methods that data were collected on day 1, 3 and 7. Please clarify.

- Similarly, figure 1 displays platelet count, the oxygenation index and the the six-point-ordinal score up to day 10 which is contrasting the statement mentioned in the methods. Please clarify.

- 43.7 % of the patients required mechanical ventilation but the duration of mechanical ventilation was 0 [0-10] ? I would suggest to the authors to double check this finding.

Discussion

- Mechanism of thrombocytopenia: In addition to the mechanisms of thrombocytopenia discussed by the authors, other factors should be probably discussed for the group of patients included in the study:

o Drug-associated thrombocytopenia: 18.5 % received linezolid and others received chloroquine phosphate.

o ECMO was required in 12 % of the patients in CRRT in 20 % : in addition to heparin requirement, both ECMO and CRRT have been associated with thrombocytopenia (ref: Panigada M, Minerva Anestesiol. 2016 Feb; 82(2):170-9 - Weingart C, Artif Organs. 2015 Sep; 39(9):765-73.

o Procalcitonin was higher in patients with thrombocytopenia and higher incidence of septic shock was seen in patients with thrombocytopenia). The role of inflammation as a cause of thrombocytopenia should be discussed.

- Platelets and respiratory function:

The authors well described that thrombocytopenia is incriminated in worsening respiratory function. However, thrombocytopenia can also be the consequence of the disease. In fact, endothelial dysfunction in patients with acute respiratory distress syndrome has been identified as a cause of thrombocytopenia.

Minor comments

- Line 69: organ dysfunction rather than organ function.

- Line 175 (median survival time in patients with thrombocytopenia: Please add the quartiles. It would be better to include this statement in the results rather than in the title of figure 2.

6. PLOS authors have the option to publish the peer review history of their article (what does this mean?). If published, this will include your full peer review and any attached files.

Reviewer #1: No

Reviewer #2: **Yes: **Anis Chaari

---

## [Author Response · Author response to Decision Letter 0]

22 Jan 2021

PLOS ONE

December 28,2020

Dear Editor and reviewers.

Thank you very much for your letter and advice on our manuscript titled “Association between thrombocytopenia and 180-day prognosis of COVID-19 patients in intensive care units: a two-center observational study” (Manuscript ID PONE-D-20-31221). All comments you given are helpful and valuable for improving our article. 

We have seriously discussed about all these comments, and made a modification on the revised manuscript. Changes made to our original manuscript are marked in red. Other supporting information files, such as telephone questionnaire and the minimal anonymous data will also be uploaded as your requested. 

With regard to the reviewer’s comments and suggestions, we reply as the followings.

We hope the revised manuscript will better suit PLoS One but are happy to consider further revisions, and we thank you for your continued interest in our research. 

Sincerely,

Zhiyong Peng, MD

Department of Critical Care Medicine

Zhongnan Hospital of Wuhan University

Editor Comments:

You should expand a little on the section on mechanism of thrombocytopenia. For instance providing an example of viruses that are associated with thrombocytopenia with known mechanism of actions. I have attached a review on this topic which should assist in identifying examples (Alonso AL, Cox D. Platelet interactions with viruses and parasites. Platelets. 2015;26(4):317-23). Furthermore, Covid-19 causes a viral sepsis which is very similar to bacterial sepsis and there have been many studies on how bacteria cause thrombocytopenia. Here is a review on this which may help you identify suitable examples (Kerrigan SW, Devine T, Fitzpatrick G, et al. Early Host Interactions That Drive the Dysregulated Response in Sepsis. Frontiers in Immunology. 2019;10(1748).

Response: Thank you for this thoughtful comment. We have added the two articles you mentioned to illustrate that platelets can interact with pathogens and are involved in the immune process (Page 20, line 210-212; Page 21, line 244-245). In addition, we also added two concrete examples of influenza A H1N1 and EMCV, which belong to the same family of single-stranded RNA viruses as SARS-CoV-2, and explained how the interaction between platelet and viruses leads to platelet activation and increased platelet consumption (Page 21-22, line 245-249). 

Review Comments

Reviewer #1: 

1. In this study, the authors try to correlate platelet counts with deterioration of respiratory function and prognosis. Such studies already exist. Major1. The authors should try and explain how this study is different to papers published already on thrombocytopenia and poor prognosis in COVID-19 (for example; ?longer follow-up)

Response: Thank you for your comment. The association between thrombocytopenia and poor prognosis in COVID-19 patients has been reported in previous studies. However, we presented new information about the prognosis of critically ill COVID-19 patients in longer follow-up period (such as 90-day and 180-day mortality) and the association between thrombocytopenia and respiratory dysfunction. We stated it in the discussion section (Page 20, line 210-217).

2. Several factors were noted to be associated with poor prognosis on univariate analysis. How important is platelet count among these？

Response: Thank you very much for your question. We did the cox regression and multivariable analysis. All three risk factors of 28-day mortality and all four risk factors of 180-day mortality are shown in following tables. The B in tables means β (regression coefficient). The β of thrombocytopenia are higher than the other risk factors. Thus, we believe that thrombocytopenia contributes more to death, and is more important than the other mortality risk factors.

. Variables in the Equation of 28-day mortality.

 Variables in the Equation of 180-day mortality.

3. The significant finding is the correlation with respiratory function. it would be useful to build upon this finding

Response: Thank you for your suggestion. We added the analysis on the relationship between different levels of platelets and oxygenation indexes at admission (See S2 Fig), and found that patients with lower platelets had lower oxygenation index. We stated it in result section (page 16, line 163). For your convenience, the figure is also presented below. The figure, combined with the time curves we provided in revised manuscript (Result section, Fig 1), may better illustrate the association between platelet count and respiratory function.

S2 Fig 1. The platelet counts and oxygenation index at admission

4. You explain on page 20, line 162...the worse respiratory function was probably related to the lower platelets. But the tendencies in survival group didn’t seem to make sense -- could you elaborate

Response: We are grateful for your thoughtful comment. The downtrends of platelet count and respiratory function were only seen in non-survival group, but not in survival group. It is because the platelet counts in most survivors were not low and their respiratory functions were not too bad. We compared the severity of disease between survivors and non-survivors and found survivors had better organ functions than non-survivors (See S3 Table), which is consistent with the point that thrombocytopenia is related with severity of COVID-19 (Lippi G, 2020). We added the sentence in revised manuscript: “But the downtrends were not seen in survival group (Fig 1c, d), which may come down to the different incidences of thrombocytopenia and severity of disease (S3 Table)” (Result section, page 16, line 167-169).

S3 Table 1. Dynamic changes of organ functions.

Variables Days Survival group Non-survival group P value

Platelet count 1 209.00 [149.75-257.00] 172.00 [121.00-223.00] <0.01

 3 186.00 [138.75-261.00] 166.00 [113.50-245.75] 0.07

 7 233.000 [160.50-293.50] 132.00 [95.25-237.50] <0.01

six-point ordinal scale 1 4.00 [3.00-4.00] 4.00 [3.00-5.00] 0.06

 3 4.00 [3.00-4.00] 4.00 [4.00-5.00] <0.01

 7 3.50 [3.00-4.00] 5.00 [4.00-5.00] <0.01

Oxygenation index 1 135.75 [89.53-238.14] 113.20 [86.90-202.80] 0.36

 3 201.30 [122.48-286.33] 151.70 [91.65-224.38] 0.03

 7 212.00 [146.82-295.84] 117.20 [82.80-182.20] <0.01

SOFA 1 3.50 [2.00-5.00] 5.00 [3.00-7.00] <0.01

 3 4.00 [3.00-5.00] 5.00 [4.00-7.00] <0.01

 7 3.00 [2.00-5.25] 5.00 [3.00-8.00] <0.01

Creatinine, μmol/L 1 64.40 [53.60-89.70] 75.50 [56.80-134.20] 0.08

 3 64.30 [49.90-77.00] 77.50 [52.85-140.45] 0.07

 7 60.40 [48.70-75.30] 68.00 [49.70-189.90] 0.11

blood urea, mmol/L 1 9.80 [6.30—15.10] 10.10 [6.20-16.44] 0.89

 3 6.20 [4.35-8.60] 9.30 []5.79-16.00] <0.01

 7 6.50 [4.50-8.40] 11.01 [6.80-20.60] <0.01

AST, IU/L

 1 31.50 [19.80-55.50] 40.00 [24.00-65.00] 0.12

 3 29.50 [18.00-54.25] 36.50 [22.75-58.50] 0.22

 7 28.00 [19.00-53.25] 29.00 [20.75-51.50] 0.67

ALT, IU/L 1 29.00 [19.00-52.00] 34.00 [20.00-47.00] 0.74

 3 33.00 [19.00-61.75] 32.50 [18.00-49.00] 0.46

 7 36.50 [20.00-75.25] 25.00 [13.50-43.50] 0.01

Lymphocyte percentage, % 1 9.90 [5.50-17.30] 6.70 [3.60-10.40] <0.01

 3 12.40 [9.70-18.10] 5.85 [3.35-10.43] <0.01

 7 15.35 [11.98-21.50] 5.40 [2.80-6.70] <0.01

Procalcitonin, mg/mL 1 0.10 [0.06-0.24] 0.32 [0.12-1.04] <0.01

 3 0.10 [0.05-0.18] 0.39 [0.14-2.10] <0.01

 7 0.09 [0.05-0.27] 0.46 [0.14-1.52] <0.01

PT, seconds 1 12.60 [11.63-13.50] 13.25 [12.18-15.33] <0.01

 3 12.7 [11.65-13.38] 14.20 [12.85-15.85 <0.01

 7 12.25 [11.50-13.20] 13.60 [12.53-15.30] <0.01

APTT, seconds

 1 31.60 [27.80-34.70] 32.00 [27.95-41.50] 0.11

 3 30.35 [26.55-35.18] 34.40 [28.35-41.55] 0.02

 7 30.00 [26.55-36.03] 35.65 [30.35-42.98] <0.01

WBC, ×109/L

 1 7.32 [5.17-10.08] 9.59 [7.02-13.40] <0.01

 3 7.03 [5.01-9.46] 9.09 [7.27-12.11] <0.01

 7 6.86 [5.60-8.87] 11.26 [7.52-13.79] <0.01

hs-CTn I, ng/mL 1 0.01 [0.01-0.03] 0.03 [0.02-0.09] <0.01

 3 0.02 [001-0.05] 0.06 [0.03-0.21] <0.01

 7 0.03 [0.01-0.09] 0.07 [0.03-0.23] 0.01

5. page 23, line 193 says, shorter ICU length of stay, which might be related to more early death events -- is it possible to confirm this

Response: Thank you very much for this comment. Patients with thrombocytopenia had shorter ICU length of stay and higher mortality than non-thrombocytopenia group in our study. And we believe that this phenomenon is due to more early death events within ICU in thrombocytopenia group. The median ICU length of stay was 12 days in the whole population, so we did a 14-day Kaplan-Meier analysis (See S4 Fig 1). The survival curve of thrombocytopenia group drops faster than the other group (P<0.01), which means the patients with thrombocytopenia suffered a higher risk of early death within ICU.

S4 Fig 1. 14-day Survival Curve in Patients with and without Thrombocytopenia.

6. page 27, line 212, Most death events occurred within 90 days, which means the COVID-19 is an acute attack to those affected -- what is meant by acute attack

Response: We appreciate and agree to your comments. We compared the mortality within 180 days and found that most death events occurred within 28 days. Thus, we considered COVID-19 as an acute attack to human health. We revised our manuscript as follow: “86.6% of death events occurred within 90 days, and 66.67% of them occurred within 28 days, which means this disease developed rapidly and led to death in a short time in high-risk patients” (Discussion section, page 20, line 206-208).

7. In the mechanism of thrombocytopenia, the role of high Von Willebrand levels is worth explaining

Response: Thank you for this helpful comment. We had added the relevant content related to high von Willebrand factor in revised manuscript: “As a result of endothelial injury, sub-endothelial collagen exposure promotes innate adhesion of platelets to collagen through von Willebrand factor, which is vital in the adhesion process and significantly increased in COVID-19 patients [32, 48]” (Discussion section, page 22, line 250-252).

8. the discussion on thrombocytopenia and respiratory function would benefit from some explanation for causation. The description here is about role of platelets rather than thrombocytopenia

Response: We appreciate for this helpful comment. We described the role of platelet in the revised manuscript (Discussion section, page 20, line 210-217). Platelets can interact with virus and bacteria via surface cell receptors, and participate in the process of pathogen clearance and lung injury. SARS-CoV-2 couldn’t be effectively cleared in patients with thrombocytopenia, and high load of SARS-CoV-2 in lung led to endothelial injury, microthrombi formation, and subsequently worsening acute lung injury. The role of interaction between platelets and endothelial cells is discussed in revised manuscript (Page 20-21, line 219-232).

9. use mortality risk factors rather than Death risk factors

Response: Thank you for your revise. We have corrected this description

Reviewer #2: 

I would like to thank the editor for giving me the opportunity to review this article. The authors reported a detailed data of a cohort of sars-cov2 patients and thoroughly investigated the prognostic value of thrombocytopenia in this group of patients.

Major Comments

1. Results: One the major findings of the study was that both thrombocytopenia and prolonged aPTT were associated with higher 28-day and 180-day mortality. This finding may suggest that patient who died would have higher incidence of disseminated intravascular coagulation (DIC). Tang N et al reported that the incidence of DIC in patients who died because of sars-cov2 pneumonia was as high as 71.4 % (J Thromb Haemost, 2020). This parameter was not investigated per se by the authors. I would advise the authors to define it and to investigate its value in predicting mortality.

Response: We really appreciate this thoughtful comment. The DIC is a clinical diagnosis, which needs to be diagnosed by combining the patient’s condition, medication and laboratory indexes and other data. We tried to define it, but we found that the incomplete clinical data and laboratory indexes make it impossible. However, we had defined the coagulation disorder using APTT and PT and found that patients died within 28 days had significant higher incidence of coagulation disorder than survivors (44.8% vs 15.6%, p<0.01). This may partly prove that coagulation disorder is associated with death in critically ill COVID-19 patients.

2. Results: Table 3 displays the data on day 2, 2 and 3 while the authors mentioned in the methods that data were collected on day 1, 3 and 7. Please clarify.

Response: Thank you. We apologize for the typo. We have corrected this error in Table 3.

3. Similarly, figure 1 displays platelet count, the oxygenation index and the six-point-ordinal score up to day 10 which is contrasting the statement mentioned in the methods. Please clarify.

Response: Thank you for pointing out this neglect. We have now modified the sentence in revised manuscript: “Platelet count, six-point ordinal scale scores and oxygenation indexes were recorded on the 1st, 3rd, 7th and 10th day” (Method section, page 4, line 90-91).

4. 43.7 % of the patients required mechanical ventilation but the duration of mechanical ventilation was 0 [0-10] ? I would suggest to the authors to double check this finding.

Response: We apologize for the typo. We checked and re-analyzed our data. We modified the duration of invasive ventilation in table 4 (Result section, Page 18-19). 

5. Discussion

-Mechanism of thrombocytopenia: In addition to the mechanisms of thrombocytopenia discussed by the authors, other factors should be probably discussed for the group of patients included in the study:

o Drug-associated thrombocytopenia: 18.5 % received linezolid and others received chloroquine phosphate.

o ECMO was required in 12 % of the patients in CRRT in 20 % : in addition to heparin requirement, both ECMO and CRRT have been associated with thrombocytopenia (ref: Panigada M, Minerva Anestesiol. 2016 Feb; 82(2):170-9 - Weingart C, Artif Organs. 2015 Sep; 39(9):765-73. 

o Procalcitonin was higher in patients with thrombocytopenia and higher incidence of septic shock was seen in patients with thrombocytopenia). The role of inflammation as a cause of thrombocytopenia should be discussed.

Response: Thank you for this helpful comment. We added other possible mechanisms including those you mentioned above: “Antiplatelet antibodies were also confirmed to be cause of vancomycin-induced thrombocytopenia [49, 50]. Other than that, the drugs such as linezolid, ribavirin, interferon, cephalosporin and chloroquine phosphate may result in drug-associated thrombocytopenia. [50-53]. ECMO and CRRT have also been reported to be associated with decrease of platelet count [54-57]. High incidences of secondary infections and septic shock were seen in our study, and the intense inflammation response will promote thrombosis and lead to thrombocytopenia as well [33, 58]” (Discussion section, page 22, line 255-261).

6. Platelets and respiratory function: 

The authors well described that thrombocytopenia is incriminated in worsening respiratory function. However, thrombocytopenia can also be the consequence of the disease. In fact, endothelial dysfunction in patients with acute respiratory distress syndrome has been identified as a cause of thrombocytopenia.

Response: Thank you for your comment. We agree with you. We have modified the discussion section, and added the sentence in revised manuscript: “Endothelial injury was reported to be directly caused by SARS-CoV-2 [26]. The endothelial injury and cytokines released from endothelial cells can also lead to platelet activation and aggregation, leading to microthrombi formation and a decrease in platelet count [26, 32], which may also result in thrombocytopenia and lung injury. Thus, the association between thrombocytopenia and respiratory dysfunction is quite intricate and cannot be explained by simple causation” (Discussion section, page 21, line 227-232). We also mentioned it in limitation section that further studies are needed to confirm their association. 

Minor comments- 

7. Line 69: organ dysfunction rather than organ function.

Response: Thank you. We have corrected it.

8. Line 175 (median survival time in patients with thrombocytopenia: Please add the quartiles. It would be better to include this statement in the results rather than in the title of figure 2.

Response: Thank you for this helpful commend. We deleted the description from the title of figure 2, and added the quartiles in the result section in the revised manuscript: “The median survival time in patients with thrombocytopenia was 17 (IQR, 7.5-180) days” (Result section, page 16, line 171).

---

## [Decision Letter · Decision Letter 1]

4 Mar 2021

Association between thrombocytopenia and 180-day prognosis of COVID-19 patients in intensive care units: a two-center observational study.

PONE-D-20-31221R1

Dear Dr. Peng,

We’re pleased to inform you that your manuscript has been judged scientifically suitable for publication and will be formally accepted for publication once it meets all outstanding technical requirements.

Kind regards,

Dermot Cox

Academic Editor

PLOS ONE

Additional Editor Comments (optional):

Reviewers' comments:

Reviewer's Responses to Questions

**Comments to the Author**

1. If the authors have adequately addressed your comments raised in a previous round of review and you feel that this manuscript is now acceptable for publication, you may indicate that here to bypass the “Comments to the Author” section, enter your conflict of interest statement in the “Confidential to Editor” section, and submit your "Accept" recommendation.

Reviewer #2: All comments have been addressed

2. Is the manuscript technically sound, and do the data support the conclusions?

Reviewer #2: Yes

3. Has the statistical analysis been performed appropriately and rigorously? 

Reviewer #2: Yes

4. Have the authors made all data underlying the findings in their manuscript fully available?

Reviewer #2: Yes

5. Is the manuscript presented in an intelligible fashion and written in standard English?

Reviewer #2: Yes

6. Review Comments to the Author

Reviewer #2: Dear editor,

The authors did substantial changes in the manuscript. I would advise for accepting the revised version.

Regards

7. PLOS authors have the option to publish the peer review history of their article (what does this mean?). If published, this will include your full peer review and any attached files.

Reviewer #2: **Yes: **Anis Chaari

---

## [Editor Report · Acceptance letter]

11 Mar 2021

PONE-D-20-31221R1 

Association between thrombocytopenia and 180-day prognosis of COVID-19 patients in intensive care units: a two-center observational study. 

Dear Dr. Peng:

I'm pleased to inform you that your manuscript has been deemed suitable for publication in PLOS ONE. Congratulations! Your manuscript is now with our production department. 

Kind regards, 

on behalf of

Dr. Dermot Cox 

Academic Editor

PLOS ONE